# Dietary Supplementation with Linseed Oil Ethyl Esters Improves Sexual Behavior and Chosen Seminal Parameters in Porcine Species

**DOI:** 10.3390/ani13081347

**Published:** 2023-04-14

**Authors:** Anna Zwyrzykowska-Wodzińska, Wiesław Bielas, Wojciech Niżański, Anna Jankowska-Mąkosa, Damian Knecht

**Affiliations:** 1Institute of Animal Breeding, Wroclaw University of Environmental and Life Sciences, 50-375 Wroclaw, Poland; 2Department of Reproduction and Clinic of Farm Animal, Wroclaw University of Environmental and Life Sciences, 50-375 Wroclaw, Poland

**Keywords:** boar semen, linseed oil ethyl esters, sperm quality

## Abstract

**Simple Summary:**

The work concerns upon the supplementation of linseed oil ethyl esters to boars in order to improve semen parameters. The study included two groups of animals each with *n* = 6 animals. The application of linseed oil ethyl esters at 3% to the complete feed mixture in the experimental group improved semen parameters. It significantly increased semen parameters in particular sperm viability (*p* < 0.001), higher semen volume (310 mL versus 216 mL, *p* < 0.001) and sperm concentration (331 versus 216 million per mL, *p* < 0.001). Higher semen quality parameters of normospermic boars may have a direct translation into prolonged use of animals, which will significantly reduce pig production costs.

**Abstract:**

Numerous studies have shown that improvements in the sperm and semen quality of males of many species can be achieved with appropriate dietary supplements added to feed or fodder. Particularly promising seems to be the inclusion of omega polyunsaturated fatty acids in the diets of males. Among other things, it has been shown that linseed oil ethyl esters (EELO can be an excellent source of omega 3 polyunsaturated fatty acids in animal diets. These compounds are more durable and resistant to oxidation, epoxidation and resinification processes, and do not exhibit toxic properties in living organisms. At present, there is a lack of data in the literature on the enrichment of boar diets with EELO. The purpose of this study was to analyze the effects of the addition of EELO to boar diets on the properties of sperm in fresh semen. The study was conducted during the summer on semen collected from 12 boars of the line 990. Linseed oil ethyl esters were administered in each feeding at a rate of 3.0% (45 mL each) in basal diets for each boar on a daily basis for 16 weeks. Ejaculates were collected manually by the gloved-hand technique, at one-week intervals for eight-week periods, from the eighth week onwards after the start of feeding. Eight ejaculates were collected from each boar, totaling 96 samples. The addition of EELO to the diets of boars caused an increase in sperm viability (*p* < 0.001), semen volume (310 mL versus 216 mL, *p* < 0.001) and sperm concentration (331 versus 216 million per mL, *p* < 0.001). Furthermore, in the experimental animals, there was a decrease in the percentage of spermatozoa exhibiting DNA fragmentation. The experimental boars also showed an increase in the percentage of gametes without apoptosis and capacitation and an increase in the percentage of viable spermatozoa not showing lipid peroxidation membranes. Consequently, EELO nutritional supplementation resulted in the improved quality of the fresh semen of boars.

## 1. Introduction

The improvement of animal production is indispensable and equivalent to the development of a market economy in competitive conditions. Owing to the quality requirements for pig producers, the success of production is primarily determined by the cost of raw material production and the quality of the product. The reproduction of pigs is one of the production phases with the largest reserves. The efficiency of reproduction in pigs resulting in healthy offspring depends on different factors, such as environment, species features, race and inter-individual variation. Breeding progress and advances in the breeding values of pig herds correlating with changes in consumer expectations and increasing husbandry requirements are being realized on a large scale in boars. Therefore, further identification of threats to their welfare and factors affecting performance (semen quality and quantity) is a constant motivation for research on animal production and the use of boars in fattening pig production. A challenge for both researchers and producers is the search for additives to support boar nutrition, affecting the metabolism and physiology of sperm production, which should translate into more efficient use of utility boars in herds. [1].

Boars should be characterized by high production value in terms of traits passed on to the pigs’ offspring and very good quality semen. Laboratory diagnostics regarding sperm quality involve macroscopic and microscopic analyses. Sperm examination includes parameters such as ejaculate volume, sperm concentration, sperm motility, membrane integrity (intact), acrosome status, apoptotic markers, color and the smell of the ejaculate [2]. Extensive biochemical studies have shown that the pig sperm surface is organized into lipid domains significantly different from those in somatic cells. Fatty acids present in semen include saturated fatty acids (SFAs), such as palmitic (16:0), stearic (18:0) and myristic (14:0) acids, and polyunsaturated fatty acids (PUFAs), such as docosapentaenoic acid (22:5 ω 3) and docosahexaenoic acid (22:6). Oleic acid (18:1 ω 9), linoleic acid (18:2 ω 6) and arachidonic acid (20:4 ω 6) are also present, but in smaller amounts [3].

Polyunsaturated fatty acids, especially DHA, have a great influence on spermatogenesis, sperm maturation, sperm quality and male reproductive maintenance in general [4].

Due to the presence of polyunsaturated fatty acids in semen, it is susceptible to oxidation when reactive oxygen species (ROS) production exceeds the buffering capacity of antioxidants. High levels of ROS have been implicated in abnormal semen parameters [5]. 

This effect can be withheld by the application of antioxidants. Supplementation with dietary PUFAs from different sources, such as fish oil [6], microalgae [7] and flaxseed oil [8,9,10], is practicable, and the positive effects on boar semen have been confirmed.

Flax (*Linum usitatissimum* L.) is undeniably one of the oldest cultivated plants. Flax is a spring plant grown on loamy and sandy loam soils [11]. Two forms of flax are cultivated today: long-stem fiber flax and branched oil flax. Oil varieties have numerous branches, their fibers are shorter and therefore more difficult to spin, but their seeds contain up to 40% oil. Oilseed varieties also have more seeds per plant, and the seeds are also larger than those of fiber varieties [12].

Flax seeds contain significant amounts of polyunsaturated fatty acids, including the valuable α-linoleic acid (C18:3, ω6), and biologically active lignans, which are used as raw materials in the chemical and pharmaceutical industries [13].

The remaining pomace after obtaining the oil is a source of proteins and can be used as animal feed as well as for human consumption. Flax, as a source of omega 3 essential fatty acids (EFAs), is an important product used in the form of whole seeds, crushed seeds and oil in the nutrition of many animal species [14,15]. The α-linolenic acid contained in linseed oil belongs to the omega-3 EFAs. Deficiency in this acid manifests in disorders of the proper functioning of the body (cell membranes, skin, the immune system, the circulatory system and the nervous system). It has a positive effect on the quality of male semen [8,9,10]. It can elongate and add new double bonds and thus convert into eicosapentaenoic acid (EPA) and docosahexaenoic acid (DHA) (Figure 1) [16].

Linseed oil ethyl esters are an excellent source of omega 3 fatty acids, especially α-linoleic acid, which represent about 50–60% of the total pool of fatty acids. Unlike linseed oil, its ethyl esters are devoid of harmful substances, such as amygdalin and linamarin, which damage liver cells. Due to the lower solubility of oxygen in esters, they are more durable and less susceptible to oxidation, epoxidation, peroxidation and gumming processes, and it seems that they can have a positive effect on the body [17,18].

At present, there are no data in the literature on the enrichment of boars’ diets with EELO. The aim of the study was to evaluate the effect of dietary linseed oil ethyl esters on the sexual behavior and quality parameters of fresh semen (SQP) of normospermic boars.

## 2. Materials and Methods

### 2.1. Experiment Location 

The experiment was conducted at the Department of Reproduction with the Clinic of Farm Animals, Wrocław University of Life Sciences. The trial was performed in summer, when the mean temperature was a minimum of 21.8 °C.

The animal study protocol was approved by the Ethics Committee of Wrocław University of Environmental and Life Sciences (protocol code: 45/2015; date of approval: April 2015) for studies involving animals.

### 2.2. Animals

In the present study, twelve Polish line 990 breed boars ranging between 20 and 26 months of age were used. The boars were purchased from the Pawłowice Experimental Plant. They were clinically healthy and had normal locomotor function. Males were trained to mount an artificial sow and donate semen on a phantom, and all had the normal semen quality criteria, i.e., >50 × 10^8^ total sperm cells per ejaculate, initial motility > 70% and containing > 70% morphologically normal spermatozoa. Males underwent the required quarantine after purchase; tested negative for brucellosis, Aujeszky’s disease and PRRS; and were dewormed against external and internal parasites and vaccinated against erysipelas and parvovirus. Blood morphology tests were conducted and confirmed the health status of the animals. The average body weight of the animals was 179.3 ± 9.19 kg (159–191 kg). The boars were maintained in an experimental piggery, in individual pens on a concrete floor with thermal and moisture isolation covered by litter with an area of 9 m^2^. The boars were kept under the same controlled environmental conditions: uniform feeding and lighting and other standard management practices used at the Department of Reproduction with the Clinic of Farm Animals.

### 2.3. Experimental Design

The feeding trial lasted for a period of 16 weeks. Boars were randomly assigned to one of two groups: the Control or the Experimental group (*n* = 6 per group), which were balanced for boar age and semen quality. Boars were fed individually with a pelleted complete feed, with a restriction of 3.0 kg feed/day per animal. Feed was provided twice a day, at 07.00 h and 13.30 h. The boars were provided ad libitum access to drinking water. All animals were fed in a dose-dependent manner with complete feed mixture for breeding boars (LIRA, Krzywiń, Poland). Table 1 shows the composition of the basal diet. In the Experimental group, each boar had standard feed that was top-dressed, according to the methodology presented by Singh et al. [9] and Yeste et al. [19]; every meal contained linseed oil ethyl esters (Omegaregen^®^–nutriceutic, FLC Pharma, Wrocław, Poland). Linseed oil ethyl esters were administered at each feeding at a rate of 3.0% (45 mL each) in basal diets to each boar on a daily basis for 16 weeks. A quantity of 100 mL of the nutriceutic contained ω acids in the form of ethyl esters: ω-3 α-linolenic acid (ALA), 58 g; ω-9 oleic acid (OA), 18 g; and ω-6 linoleic acid (LA), 16 g. The diets in both of the studied groups were isocaloric and isonitrogenous, based on the nutritional requirements recommended by the National Research Council NCR for breeding boars. 

### 2.4. Semen Collection and Processing

Ejaculates were collected manually by the gloved-hand technique, at one-week intervals for eight-week periods, from the eighth week onwards after the start of feeding. Sampling was performed in the morning before feeding at about 6.00 am.

Eight ejaculates were collected from each boar, totaling 96 samples. Ejaculates from the boars were collected into a thermal cup with a plastic bag. The viscose filter enabled the separation and rejection of the gel fraction. The semen was collected by a trained technician. Only sperm-rich fractions of the ejaculates were collected, while pre-sperm fractions were discarded. The freshly collected ejaculates were immediately transferred to the laboratory (the lab is adjacent to the experimental piggery, so the semen transfer was not influenced by environmental factors), where they underwent standard laboratory assessment, including determination of the following traits: ejaculate volume (mL); sperm concentration (million per mL); and sperm motility, as the percentage of sperm with progressive movement. Ejaculate volume was determined after the gel fraction was discarded. Sperm concentration in the ejaculate was determined photometrically using a SpermaCue Porcine photometer (Minitube, GmbH, Tiefenbach, Germany) at a light length of 0.7 µm [20]. The percentage of spermatozoa with normal progressive movement for the total number of sperm visible in the field of view of the phase-contrast microscope was estimated in a drop of fresh semen at 400× magnification. In the laboratory, the preparation of smears for subsequent staining with Giemsa stain [21] and routine evaluation of sperm morphology (magnification 1250×) were also performed. Sperm morphologies were examined for primary and secondary defects in morphology. Primary defects in morphology included proximal droplet heads, anomalies, acrosome damage, midpiece damage and tail coils. Secondary defects in morphology included distal droplets, bent tails, tail-less heads and tail loops [22]. 

### 2.5. Sexual Behavior of Boars

The boars’ sexual activity was assessed, based on the time needed to induce successive copulatory reflexes during the period of semen collection. Boars were not exposed to estrus sows during semen collection. The following parameters of the boars’ sexual activity were measured: reaction time, as the time from entering the collection room until mounting the phantom; and false mounts (numbers), which were counted as mounting the dummy but dismounting before semen donation by the boar (mounts with no collection). The time required to induce individual sexual reflexes and their durations were determined using a stopwatch, to within one second [9,20].

### 2.6. Sperm Kinetics and Cytometry Analysis

Sperm kinetics studies were performed by CASA, as reported by Bielas et al. (2017) [23]. Ten sperm kinematic parameters—MOT, percentage of motile spermatozoa; PMOT, percentage of progressively motile spermatozoa; RAPID, subpopulation of rapid cells; VAP, average path velocity; VSL, straight-line velocity; VCL, curvilinear velocity; ALH, amplitude of lateral head displacement; BCF, beat cross frequency; STR, straightness; and LIN, linearity—were analyzed in fresh semen by CASA using the Hamilton Thorne Sperm Analyzer IVOS version 12.2L (Hamilton Thorne Biosciences, Beverly, MA, USA) under 1.89 × 10 magnification. A 3 mL aliquot of semen was placed in a Leja4 analysis chamber (Leja, Nieuw-Vannep, The Netherlands) at 35 °C and evaluated. The settings of the IVOS were as follows: frames acquired, 45; frame rate, 60 Hz; minimum cell contrast, 46; minimum cell size, 7; straightness threshold, 45%; path velocity threshold, 45 μ/s; path velocity cutoff, 20 μ/s; straight-line velocity cutoff, 5 μ/s; head size non-motile, 7; head intensity non-motile, 50; static head size, 0.65–4.90; static head intensity, 0.50–2.50; and static elongation, 0–87. Six fields randomly selected by a computer were analyzed for each semen sample. In each sample, 800–1000 sperm cells were evaluated. In order to select ejaculates, the following minimum standard values for CASA parameters were established: PMOT%: (VAP > 45 µm/s and STR > 45%); and RAPID%: (VAP > 45 µm/s). Flow cytometric analyses for cell viability, acrosome and plasma membrane integrity, apoptotic and capacitation change, membrane lipid peroxidation and sperm chromatin structure were performed on fresh samples of semen.

### 2.7. Statistical Analysis

The collected results were statistically processed using Statistica v.13.3 (TIBCO Software Inc., Palo Alto, CA, USA). The data were analyzed for normal distribution using the Shapiro–Wilk test. In order to check the homogeneity of variance, the Levene test was used. The Mann–Whitney *U* test was used to analyze the statistical differences between the means for the control and treated animals. Data are presented as mean values with SEMs, and the significance level was set at *p* < 0.001.

## 3. Results

### 3.1. Sexual Behavior and Semen Quality Parameters of Fresh Semen

The effects of 16-week dietary supplementation with linseed oil ethyl esters on the sexual behavior and quality parameters of fresh boar semen are shown in Table 2. Sexual behavior and semen quality parameters at the fresh stage were improved (*p* < 0.001) in the Experimental group as compared to the Control group (Table 2.). Reaction time was less, and the number of false mounts was lower (*p* < 0.001) in the Experimental group as compared to the Control group. Higher semen volume (*p* < 0.001) (310 mL versus 216 mL) was recorded in the Experimental group. Sperm concentration (331 versus 216 million per mL) was also higher (*p* < 0.001) in the Experimental group. Similarly, sperm motility was higher (*p* < 0.001) in the Experimental group, while the obtained spermiogram results, such as morphological primary and secondary defects, were higher (*p* < 0.001) in the Control group.

### 3.2. CASA and Flow Cytometric Assessment of Spermatozoa in Fresh Semen

#### 3.2.1. Sperm Kinetic Parameters

The effects of boar dietary supplementation with linseed oil ethyl esters on the motion characteristics assessed by a computer-assisted semen analyzer (CASA) of fresh semen are presented in Table 3. The sperm motility characteristics of the fresh semen of the boars in the Experimental group, such as MOT, PMOT and RAPID, and the kinetic properties of the sperm, such as VAP, VSL, VCL, ALH and BCF, were improved (*p* < 0.001) after dietary supplementation with linseed oil ethyl esters.

#### 3.2.2. Flow Cytometric Assessment of Fresh Semen

The effects of boar dietary supplementation with linseed oil ethyl esters on values of fresh sperm viability, acrosome and plasma membrane integrity, apoptotic changes, membrane lipid peroxidation and sperm chromatin structural assays are presented in Table 4 or in Figure 2, Figure 3, Figure 4, Figure 5, Figure 6, Figure 7, Figure 8, Figure 9, Figure 10 and Figure 11. Supplementation of the boars’ diet with linseed oil ethyl esters (*p* < 0.001) improved sperm viability as well as acrosome and plasma membrane integrity. Furthermore, the boars from the Experimental group had semen with (*p* < 0.001) increased percentages of sperm without destabilized membranes and apoptotic changes, as well as semen with (*p* < 0.001) decreased percentages of sperm with membrane lipid peroxidation and DNA fragmentation.

## 4. Discussion

The aim of the present study was to assess the effects of dietary linseed oil ethyl esters on the sexual behavior and quality parameters of fresh semen (SQP) of normospermic boars.

The preparation applied in the Experimental group consisted of linseed oil ethyl esters: ω-3 α-linolenic acid (ALA), ω-9 oleic acid (OA) and ω-6 linoleic acid (LA). Linoleic acid (LA) competes with ALA for enzymes converting alpha-linolenic acid to EPA and DHA. Easily bioavailable additives affect the animal’s body faster, which, by design, should translate into better production parameters in a shorter period of use compared to less readily bioavailable substances [24].

Linseed oil ethyl esters are characterized by significantly increased bioavailability and can be easily absorbed and incorporated into various types of blood lipid fractions. This is due to their undeveloped molecular structure and increased kinetics of free acid release, which contributes to their faster digestion. Due to significantly reduced oxygen solubility, ethyl esters are characterized by much higher durability and stability compared to linseed oil and are more resistant to oxidation, peroxidation and polymerization processes. The method of obtaining ethyl esters of polyunsaturated fatty acids is covered by patent protection [25,26].

The stability of n-3 fatty acids in flax oil during storage is moderate due to their natural auto-oxidation processes as well as the degradation of other naturally occurring compounds, i.e., cyclolinopeptides [27], which have a proven effect on feed palatability (the release of cyclolinopeptides is responsible for bitter taste). During linseed oil ethyl ester production, cyclolinopetides are eliminated from the oil [28].

Fatty acids are an important energy source. They are part of the cell membrane, metabolic substrates in many biochemical pathways and cell-signaling molecules, and they play a crucial role as immune modulators. The preventive action of fatty acids in the intestine can be correlated with their inhibitory effects on the over-release of intestinal inflammatory mediators, especially proinflammatory cytokines [29,30,31].

Fatty acids as feed additives are responsible for numerous functions. They increase pig productivity by improving the quality of intramuscular fat in pig fattening; the availability of fatty acids can influence tissue composition and exert effects on metabolism, providing adequate energy concentrations in feed, which further prevents excessive weight loss in sows due to lactation The addition of fat to feed increases its cost, but the expenses incurred are balanced by lower consumption, higher daily gains and shorter fattening times [32,33,34,35]. By deliberately supplementing feed mixtures with additives rich in PUFAs, it is possible to obtain raw meat and fat with functional food characteristics. Polyunsaturated fatty acids—especially ALA, DHA and EPA—can be classified as nutraceuticals with health-promoting benefits [36].

Vegetable oils are also a good source of n-3 family acids, and when used as components of feed mixtures they cause an increase in the contents of these acids in the lipids of the carcasses and a decrease in the contents of saturated acids. Studies conducted in recent years have confirmed the possibility of modifying the dietary qualities of pork meat by using fattening corn grain and feed mixtures lubricated with canola, flaxseed, sunflower or soybean oil. The use of the above-mentioned feed additives resulted in significant increases in the contents of PUFAs, especially of the n-3 family, in the longissimus muscles of fattening pigs. Also changed was the ratio of acids of the n-6 and n-3 families: a reduction in the ratio between these acid families is beneficial due to the proven health-promoting effects [37,38,39].

This is the first report of the application effects of EELO on boar semen quality which shows its positive effects on reproduction parameters. Previous data have [6,7,8,9,10] shown that the application of different sources of PUFAs in boar diets significantly increased semen volume, total sperm pen ejaculate and sperm concentration, in addition to reducing reaction time. This may be due to the ability of PUFAs to affect many factors which are related to the synthesis and metabolism of essential reproductive hormones, such as testosterone and prostaglandin [40]. 

Many researchers [6,7,8,9,10,41,42,43] have reported an increase in the total number of sperm in boars consuming supplements containing omega-3 fatty acids. What is more, they indicated that the application of dietary supplements increased the duration of ejaculation. On the other hand, when Murphy et al. (2017) [44] and Castellano et al. (2010) [45] used different fish oils and algae extracts rich in PUFAs, no effect on male sexual behavior was observed.

Bielas et al. (2013) [42], Singh et al. (2021) [8] and Singh et al. (2023) [10] reported that supplementation with linseed meal or oil in boars’ feed improved the motility of sperm in terms of curvilinear velocity, linearity and amplitude of lateral head displacement. 

This study demonstrated a positive effect of EELO supplementation on the quality characteristics of fresh boar sperm: viability, acrosome and plasma membrane integrity, apoptotic and capacitation changes, membrane lipid peroxidation and sperm chromatin structure. These results are consistent especially with those of previous studies conducted on linseed oil addition in boar diets [8,9,10,13]. Buhr et al.’s (2010) [46] results indicated that dietary linseed supplementation increased the fluidity of boar sperm plasma membranes. 

The observations and confirmations of other results concerning improved sperm quality due to PUFAs from EELO and linseed oil with respect to selected sperm parameters may have been due to a lowering of the degree of lipid oxidation in the spermatozoa [8,9,10,40].

In this study, all fresh semen quality parameters and CASA attributes (except linearity) were significantly improved in boars supplemented with EELO in their feed. Although there are conflicting reports on the effects of n-3 PUFAs in the diet on the semen properties of males of different species, it seems that supplementation with these compounds affects the cholesterol ratio in sperm cell membranes and thus the expression of receptors and the synthesis of sex hormones [47]. In addition, a diet rich in n-3 PUFAs (as in the case with EELO supplementation) may increase the fluidity of the sperm cell membrane and thus improve the quality of preserved semen [48].

Seasonal infertility can have a serious impact on productivity. There are ways to overcome the effects of summer heat. Declines in production parameters during the summer and early fall are observed worldwide. Seasonal infertility results from a combination of the effects of the length of the light day (a presumed residual effect of ancestral wildlife) and high temperatures. The effects are most pronounced in females, but high temperature can also affect semen quality and libido in boars. There are also clear seasonal effects on boar reproductive physiology, both in terms of semen production and libido. Boar ejaculates contain a significantly higher number of spermatozoa between September and February than between March and August. When the natural photophase was experimentally reversed between April and September, there was a corresponding increase in sperm production. Changes in photophase affect the production of sperm cells, while high ambient temperatures of 29 °C or more have a direct destructive effect on germ cells. It is therefore possible that the apparent onset of seasonal infertility in April, as measured by sow birth rates, is partly due to the natural decline in boar fertility at this time. From the end of a boar’s exposure to heat stress, at least five weeks must elapse before sperm motility returns to normal. Hot summer weather can therefore potentially cause boar infertility until early October, thus exacerbating any seasonal effects occurring in sows at that time. In addition, boar libido decreases in summer and increases in winter [49,50,51,52,53].

It is stated that, during summer months, sperm quality is lower. Decreased semen quality and fertility are associated with increased photoperiods [1,54,55] and high temperatures, which can cause heat stress [9,56]. Although our study was conducted during the summer period, it did not result in a decrease in the semen quality of the boars in the Experimental group. Therefore, it seems that the addition of linseed oil ethyl esters to the diets of boars may inhibit the effect of the photoperiod and counteract the seasonal decrease in the quality of boar semen in the summer. Therefore, the use of linseed oil ethyl esters seems to have great potential in obtaining better quality semen in the summer season. 

## 5. Conclusions

Boars supplemented with linseed oil ethyl esters showed significant improvement in seminal attributes. The supplementation of boar diets with linseed oil ethyl esters especially improved sperm viability (*p* < 0.001), semen volume (310 mL versus 216 mL, *p* < 0.001) and sperm concentration (331 versus 216 million per mL, *p* < 0.001). The analysis clearly shows that the undoubted result of this work is the improvement of semen parameters, especially sperm concentration. Furthermore, in the experimental animals there was a decrease in the percentage of spermatozoa exhibiting DNA fragmentation. The experimental boars also showed an increase in the percentage of gametes without apoptosis and capacitation and an increase in the percentage of viable spermatozoa not exhibiting lipid peroxidation membranes. This translates significantly into the number of portions from the ejaculate, resulting in a significantly better economic balance while maintaining standards for the number of sperm in semen doses, and could be utilized in production. Consequently, EELO nutritional supplementation resulted in improved fresh semen quality of boars.

## Figures and Tables

**Figure 1 animals-13-01347-f001:**
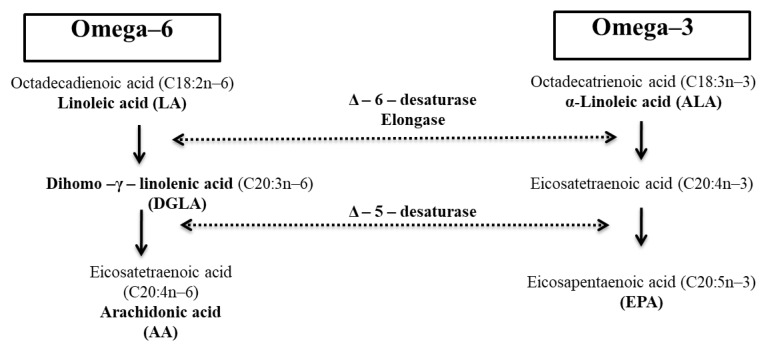
Simplified scheme of essential fatty acid metabolism.

**Figure 2 animals-13-01347-f002:**
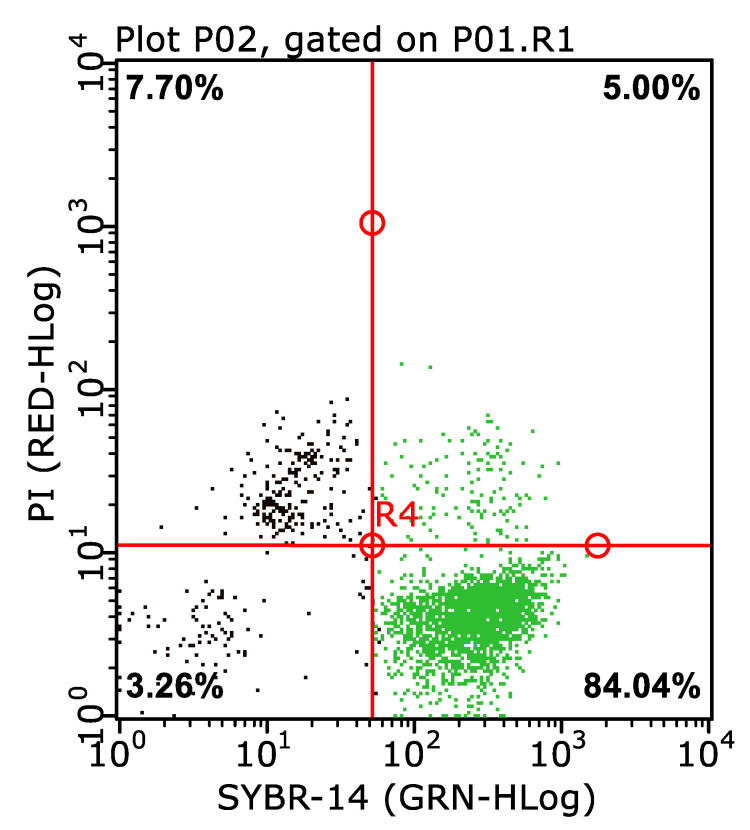
An example of a dot plot with live spermatozoa indicated by green dots in the Experimental group in the lower right quadrant (84.04%): SYBR+PI- (SYBR, SYBR-14; PI, propidium iodide).

**Figure 3 animals-13-01347-f003:**
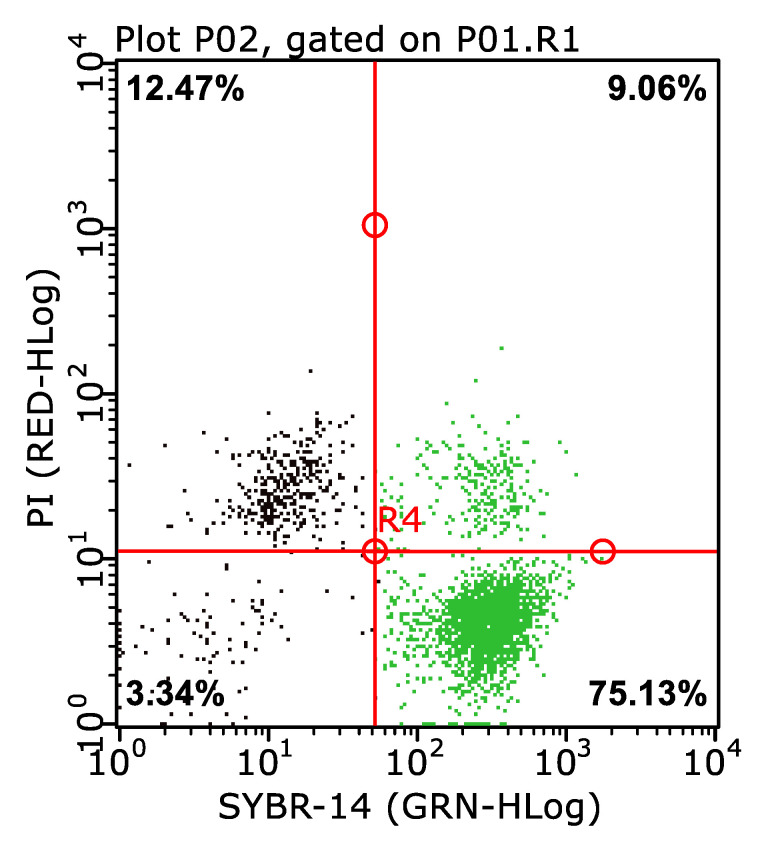
An example of a dot plot of live spermatozoa indicated by green dots in the Control group in the lower right quadrant (75.13%): SYBR+PI- (SYBR, SYBR-14; PI, propidium iodide).

**Figure 4 animals-13-01347-f004:**
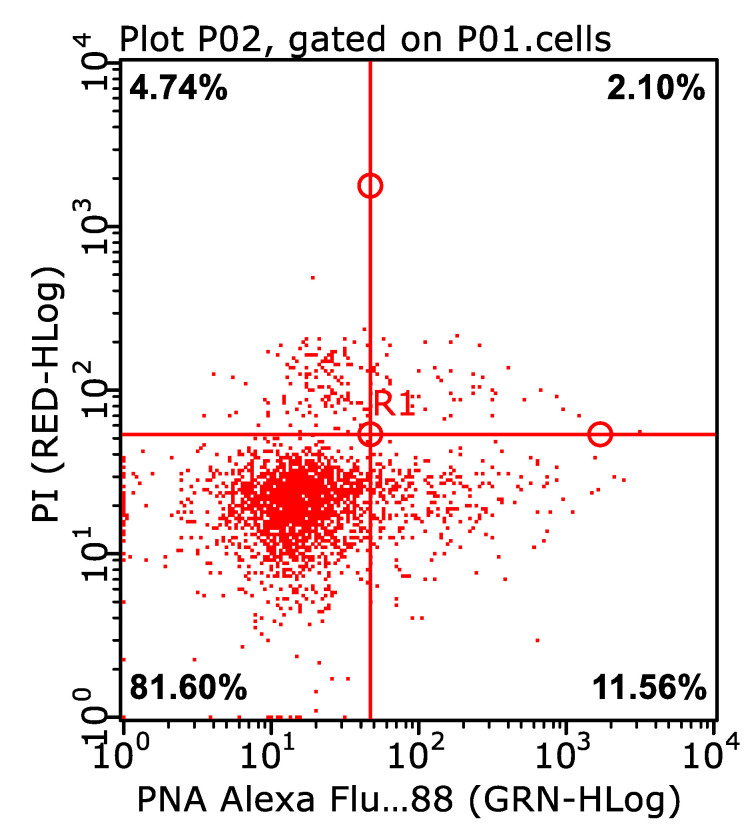
An example of a dot plot with live spermatozoa with intact acrosomes in the Experimental group in the lower left quadrant (81.6%): PNA- PI- (PNA, PNA Alexa Fluor 488–lectin from *Arachis hypogaea*; PI, propidium iodide).

**Figure 5 animals-13-01347-f005:**
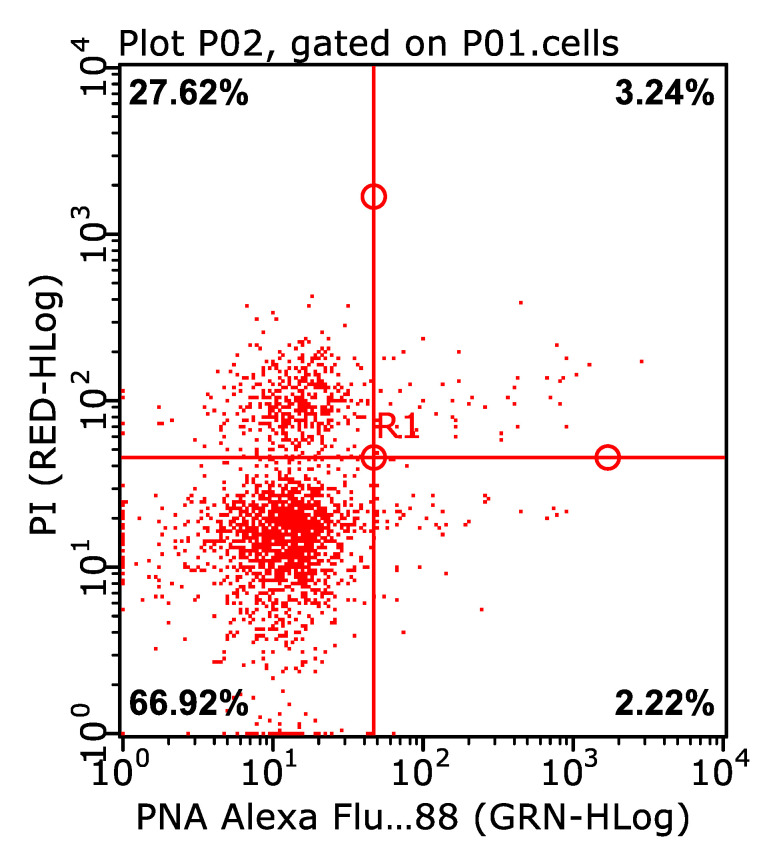
An example of a dot plot with live spermatozoa with intact acrosomes in the Control group in the lower left quadrant (86.92%): PNA-PI- (PNA, PNA Alexa Fluor 488–lectin from *Arachis hypogaea*; PI, propidium iodide).

**Figure 6 animals-13-01347-f006:**
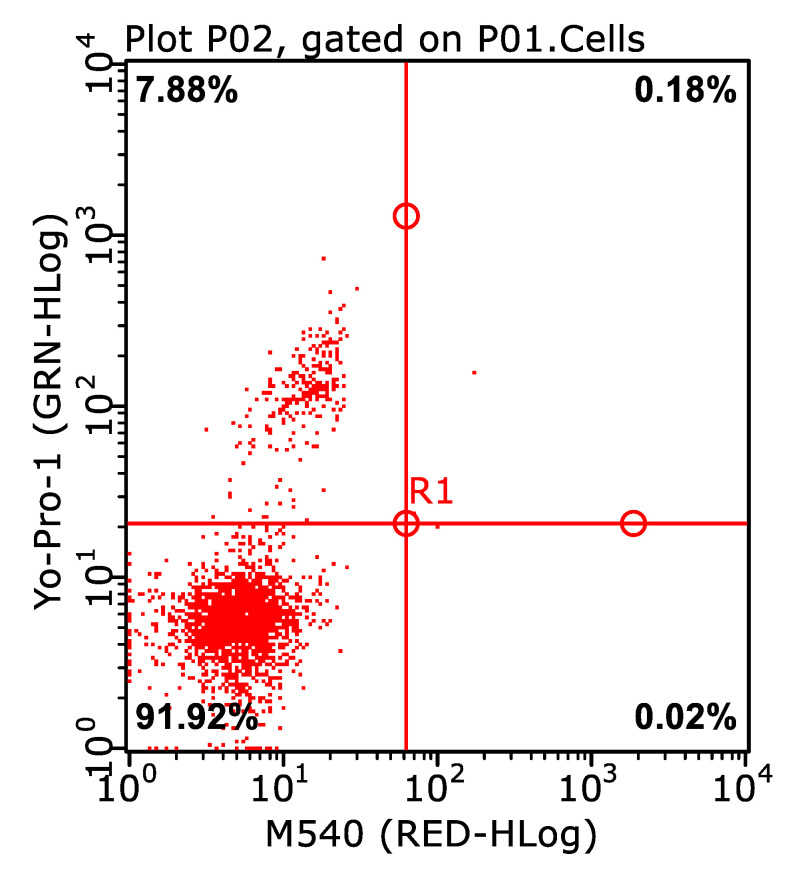
An example of a dot plot with live spermatozoa without destabilized membranes in the Experimental group in the lower left quadrant (91.82%): Mer-/Yo-Pro-1- (Merocyanine 540 (M-540) assessing capacitation-related membrane destabilization; YO-PRO-1 assay for apoptosis evaluation).

**Figure 7 animals-13-01347-f007:**
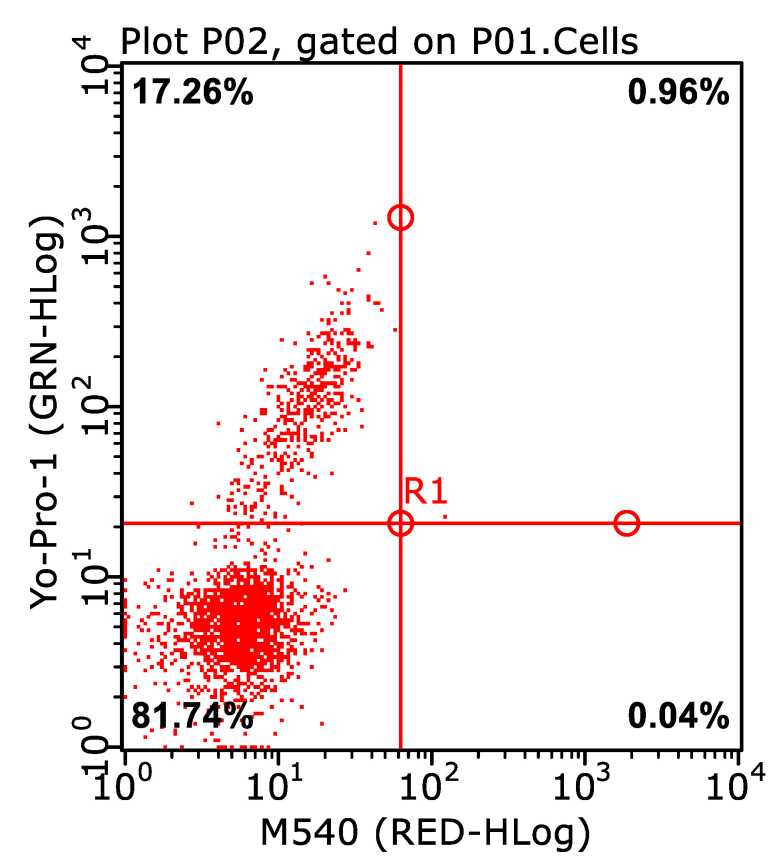
An example of a dot plot with live spermatozoa without destabilized membranes in the Control group in the lower left quadrant (81.74%): Mer-/Yo-Pro-1- (Merocyanine 540 (M-540) assessing capacitation-related membrane destabilization; YO-PRO-1 assay for apoptosis evaluation).

**Figure 8 animals-13-01347-f008:**
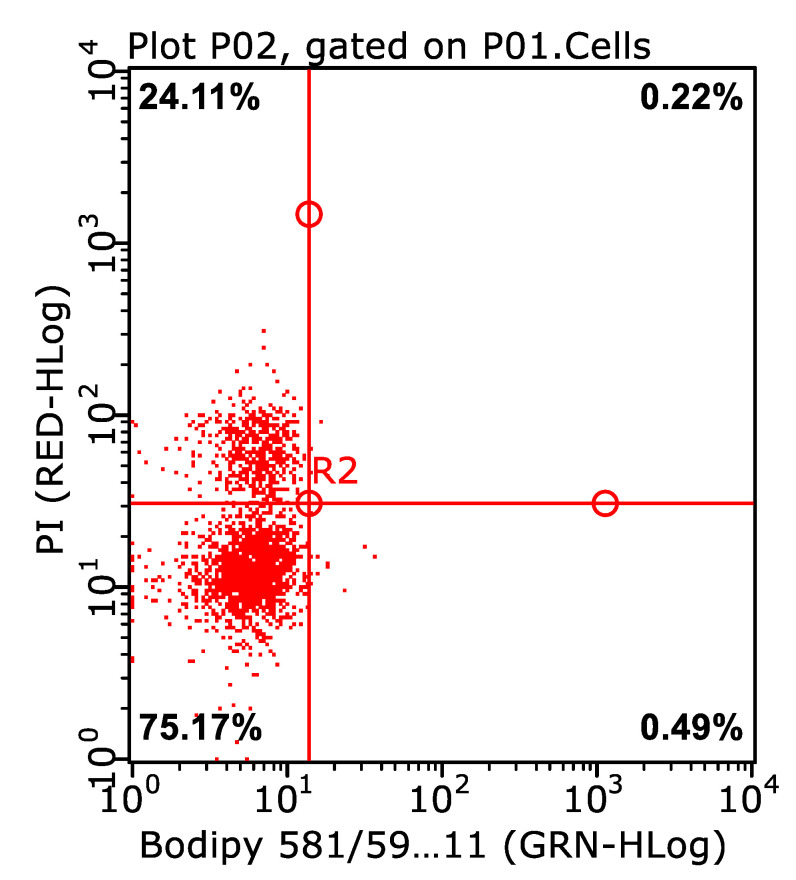
An example of a dot plot with live spermatozoa without lipid peroxidation in the Experimental group in the lower left quadrant (75.17%): PI-/BODIPY- (PI, propidium iodide; BODIPY, fluorescent membrane probe).

**Figure 9 animals-13-01347-f009:**
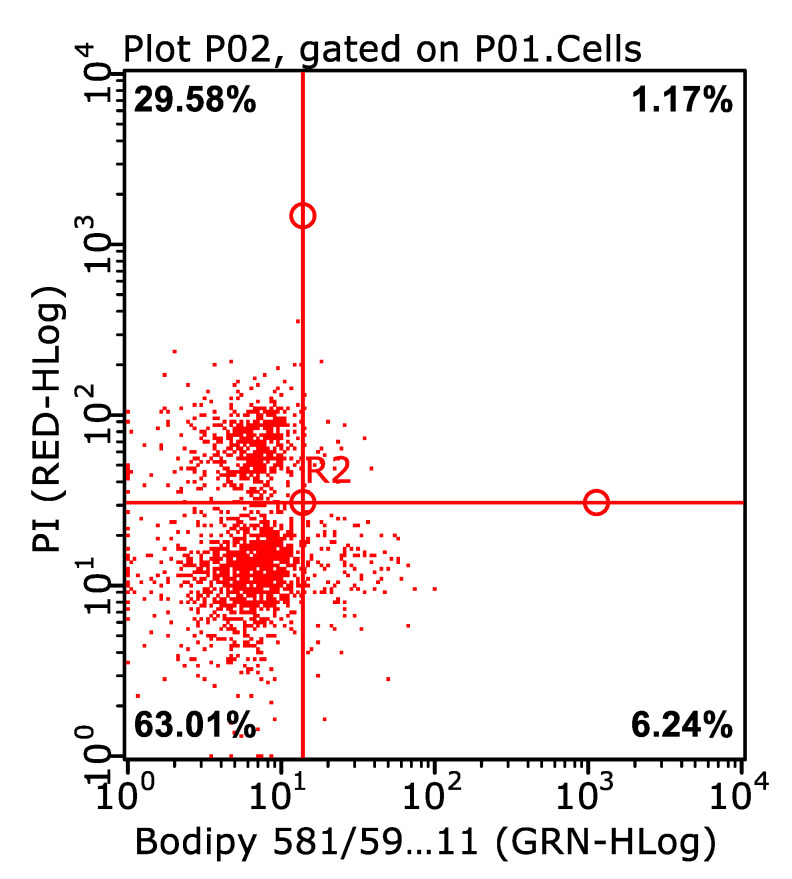
An example of a dot plot with live spermatozoa without lipid peroxidation in the Control group in the lower left quadrant (63.01%): PI-/BODIPY- (PI, propidium iodide; BODIPY, fluorescent membrane probe).

**Figure 10 animals-13-01347-f010:**
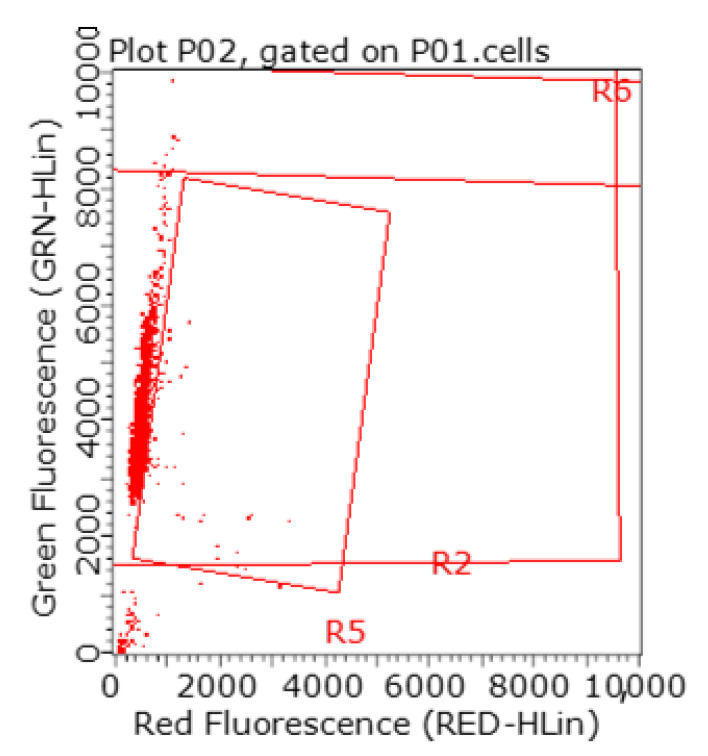
An example of a dot plot with 3.39 percent of spermatozoa with DNA fragmentation in the red rectangle in the Experimental group, determined according to the DNA fragmentation index (% DFI).

**Figure 11 animals-13-01347-f011:**
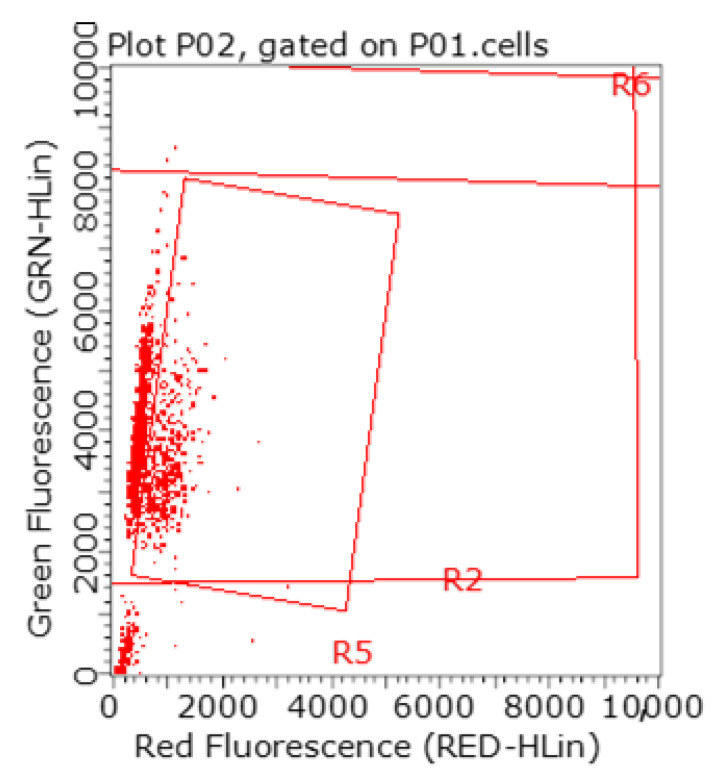
An example of a dot plot with 29.85 percent of spermatozoa with DNA fragmentation in the red rectangle in the Control group, determined according to the DNA fragmentation index (% DFI).

**Table 1 animals-13-01347-t001:** Ingredient composition (% wt/wt) of the basal diet.

Composition	
Cereal grains: barley, triticale and rye	60.8%
Wheat bran	15%
Extracted rapeseed meal	10%
Extracted soybean meal	10%
Animal fat	2%
Calcium carbonate	1.5%
Sodium chloride	0.4%
Monocalcium phosphate	0.2%
Unactivated charcoal and yeast and its parts rich in oligosacchrides of mannans and glucans	0.1%
Metabolic energy	12.50 MJ
Total protein	16%
Crude fat	4.75%
Crude fiber	5.20%
Crude ash	6%
Lys (3c *)	1%
Met (3c *)	0.36%
Met+Cys	0.65%
Tre (3c *)	0.68%
Try (3c *)	0.2%
Calcium	0.81%
Calcium phyto	0.1%
Phosphorus	0.6%
Phosphorus phyto	0.13%
Sodium	0.18
Vitamin A (E672) (3a *)	12,000 j.m.
Vitamin D (E671) (3a *)	2000 j.m
Vitamin E (3a700)	240.0 mg
Vitamin K (3a *)	5.0 mg
Vitamin C (3a *)	500.0 mg
Vitamin B1 (3a *)	2.0 mg
Vitamin B2 (3a *)	7.0 mg
Vitamin B6 (3a *)	4.0 mg
Vitamin B12 (3a *)	50.0 mcg
Biotin (3a *)	400.0 mcg
Pantothenic acid (3a *)	15.0 mg
Folic acid (3a *)	4.0 mg
Niacin (3a *)	30.0 mg
Choline (3a *)	1380.0 mg
Mn (E5) (3b *)	92.0 mg
I (E2) (3b *)	3.6 mg
Cu (E4) (3b *)	20.0 mg
Fe (E1) (3b *)	154.0 mg
Zn (E6) (3b *)	140.0 mg
Zn Chelate (E6) (3b *)	7.5 mg
Se (E8) (3b *)	400.0 mcg

Note. * Categories and functional groups of feed additives. 1a-preservatives. 1b-antioxidants. 1i-anti-caking agent. 1j-acidity regulator. 2a-colorings. 2b-flavoring substances. 3c-amino acids and their salts. 4a-digestibility enhancers. 3d-urea and its derivatives. 5-coccidiostats and histomonostats.

**Table 2 animals-13-01347-t002:** Comparison of the sexual behavior and quality parameters of fresh semen of boars fed diets top-dressed with linseed oil ethyl esters (mean, n = 6/group, SEM, *p*-value).

Parameters	Control $\bar{x}$	Experimental $\bar{x}$	SEM	*p*-Value
Reaction time (s)	236 ^A^	131.88 ^B^	6.08	0.001
False mounts (numbers)	2.08 ^A^	1.13 ^B^	0.09	0.001
Semen volume (mL)	216.7 ^A^	310.0 ^B^	6.05	0.001
Sperm concentration (million per mL)	264.8 ^A^	331.0 ^B^	4.68	0.001
Sperm motility (%)	74.0 ^A^	77.18 ^B^	0.52	0.001
Primary defects in morphology (%)	9.3 ^A^	5.0 ^B^	0.29	0.001
Secondary defects (%)	4.3 ^A^	2.6 ^B^	0.18	0.001

^A,B^ Different superscripts within a row indicate significant differences (*p* < 0.001).

**Table 3 animals-13-01347-t003:** Comparison of the sperm motility characteristics, assessed by a computer-assisted semen analyzer (CASA), of fresh semen of boars fed diets top-dressed with linseed oil ethyl esters (mean, n = 6/group, SEM, *p*-value).

Parameters	Control $\bar{x}$	Experimental $\bar{x}$	SEM	*p*-Value
MOT (%)	76.3 ^A^	85.3 ^B^	0.96	0.001
PMOT (%)	34.4 ^A^	49.1 ^B^	1.21	0.001
RAPID (%)	61.9 ^A^	77.2 ^B^	1.36	0.001
VAP (µm/s)	88.4 ^A^	117.28 ^B^	1.64	0.001
VSL (µm/s)	55.0 ^A^	70.65 ^B^	1.52	0.001
VCL (µm/s)	172.5 ^A^	208.2 ^B^	3.99	0.001
ALH (µm)	8.2 ^A^	10.3 ^B^	0.17	0.001
BCF (Hz)	27.6 ^A^	35.4 ^B^	0.61	0.001
STR (%)	60.6 ^A^	67.4 ^B^	0.70	0.001
LIN (%)	32.7 ^A^	32.1 ^B^	0.46	0.05

^A,B^ Different superscripts within a row indicate significant differences (*p* < 0.001). Characteristics assessed by CASA: MOT, percentage of motile spermatozoa; PMOT, percentage of progressively motile spermatozoa; RAPID, subpopulation of rapid cells; VAP, average path velocity; VSL, straight-line velocity; VCL, curvilinear velocity; ALH, amplitude of lateral head displacement; BCF, beat cross frequency; STR, straightness; LIN, linearity. Data are presented as means ± SEMs, and the significance level was set at *p* < 0.001.

**Table 4 animals-13-01347-t004:** Flow cytometric comparison of quality characteristics of fresh sperm: viability, acrosome and plasma membrane integrity, apoptotic and capacitation changes, membrane lipid peroxidation and sperm chromatin structure in boars fed diets top-dressed with linseed oil ethyl esters (mean, n = 6/group, SEM, *p*-value).

Characteristics	Control $\bar{x}$	Experimental $\bar{x}$	SEM	*p*-Value
Live spermatozoa (%)	90.3 ^A^	96.5 ^B^	0.44	0.001
Live spermatozoa with intact acrosomes (%)	81.8 ^A^	87.0 ^B^	0.44	0.001
Live spermatozoa without destabilized membranes (%)	76.2 ^A^	84.7 ^B^	0.72	0.001
Live spermatozoa without lipid peroxidation (%)	75.2 ^A^	84.4 ^B^	0.67	0.001
Spermatozoa with DNA fragmentation (%)	0.69 ^A^	0.29 ^B^	0.05	0.001

^A,B^ Different superscripts within a row indicate significant differences (*p* < 0.001). Characteristics assessed by flow cytometry: live spermatozoa: SYBR+PI- (SYBR, SYBR-14; PI, propidium iodide); live spermatozoa with intact acrosome: PNA-PI- (PNA, PNA Alexa Fluor 488–lectin from *Arachis hypogaea*; PI, propidium iodide); live spermatozoa without destabilized membranes: Mer-/Yo-Pro-1- (Merocyanine 540 (M-540) assessing capacitation-related membrane destabilization; YO-PRO-1 assay for apoptosis evaluation); live spermatozoa without lipid peroxidation: PI-/BODIPY (PI, propidium iodide; BODIPY, fluorescent membrane probe); percentage of spermatozoa with DNA fragmentation: DNA fragmentation index (% DFI).

## Data Availability

Research data is available from authors.

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
