# Peer review of "Dietary Supplementation with Linseed Oil Ethyl Esters Improves Sexual Behavior and Chosen Seminal Parameters in Porcine Species"

_animals, 2023, doi:10.3390/ani13081347_

Round 1

Reviewer 1 Report

Dear authors,

I congratulate you on your work, however, it requires many modifications to be accepted. Please review the statistics and if possible, could I receive the results?

Line 26: 265 millions and no 216.

The beginning of the introduction is not fluid, it needs more connection between them (line 35 – 55).

Line 107 change 108 for 108??

Tabla 1: explain primary defects and secondary defects please.

How was the statistic done? The SD is greater than the mean difference in sperm motility. When doing the statistics, has n=6 been considered in each group, or each spermatozoon or field evaluated separately?

In table 2 and 3 the same question as in table 1 about the dispersion of the data. There can be no significant differences in DNA fragmentation with these means and spreads.

Author Response

Reviewer 1:

Line 26: 265 millions and no 216.

According to review line 26 was changed

The beginning of the introduction is not fluid, it needs more connection between them (line 35 – 55).

Improved according to review.

Line 107 change 108 for 108??

According to review line 107 was changed

Table 1: explain primary defects and secondary defects please.

Primary defects of morphology are: proximal droplet head, anomalies, acrosome damaged, midpice damage, coil tail

Secondary defects of morphology are: distal droplet, bent tail, tail-less head, tail-loop

How was the statistic done? The SD is greater than the mean difference in sperm motility. When doing the statistics, has n=6 been considered in each group, or each spermatozoon or field evaluated separately?

Boars within six (n=6) in group were been considered as mean value, but in 8 samplings. Ultimately 48 samples per each group were evaluated. Is not true that the SD is greater than the mean difference in sperm motility. Sperm motility (%) for control group:74,0±7,97 A, for experimental group: 77,18±4,72 B, SEM:0,52 and p value: 0,001 Straight from Statistica table exported below.

Variable

Aggregate results Descriptive statistics (Esters)

Add

n

Mean

Minimum

Maximum

SD

Subjective sperm motility (%)

None

48

74,02083

65,00000

85,00000

4,974358

Subjective sperm motility (%)

EELO

48

77,18750

70,00000

85,00000

4,716004

In table 2 and 3 the same question as in table 1 about the dispersion of the data. There can be no significant differences in DNA fragmentation with these means and spreads.

As previously stated boars within six (n=6) in group were been considered as mean value, but in 8 samplings. Ultimately 48 samples per each group were evaluated. Straight from Statistica table exported below.

Variable

Aggregate results Descriptive statistics (Esters)

Add

n

Mean

Minimum

Maximum

SD

SCSA

None

48

0,688306

0,082305

3,474903

0,623932

SCSA

EELO

48

0,289458

0,010000

0,990000

0,208068

Reviewer 2 Report

Review Reports

Journal: Animals  

Manuscript No: animals-2329000

Manuscript Title: Linseed oil ethyl esters in boar’s diet improved sexual behavior and fresh semen quality

In general, in this paper, the authors try to address the effect of Linseed oil ethyl esters on semen quality profiles in boars. The results of the present study will help to understand the beneficial effect of Linseed oil ethyl esters supplementation on semen production and its quality profiles in porcine species in particular and livestock species in general. This investigation will be utilized in human beings to protect their fertilizing efficiency by use of the Linseed oil ethyl esters. The aim or idea of the study is good; but the studied parameters were not sufficient to prove this EELO effective in improvement of the semen production and quality profiles in porcine species. The research work is not a complete one and lacking in many aspects.

However, the present study may be considered after answering of all the following queries and re-assessing the experimental parameters.

Ø  On what basis you selected the dose of Linseed oil ethyl esters (3.0%; 45 mL) and duration of supplementation of 16 weeks? Explain

Ø  You have studied only one dose of EELO and it showed beneficial effect. What about the optimal dose?. Therefore, different doses at minimum difference between doses is to be studied to select suitable dose, which should have maxium effect on the reproductive parameters.

Ø  Have you conducted any comparison study between Linseed oil ethyl esters and Linseed oil on semen production and its quality parameters? On what basis you are claiming the Linseed oil ethyl esters treatment is better than Linseed oil?  

Ø  In this experiment, at least three groups must be included like control (without oil or its ethyl esters), with oil and with oil ethyl esters; then only this experiment will be complete one.  

Ø  You conducted this experiment in summer season and mentioned that mean temperature was minimum 21.8℃. This temperature is fall within thermo-neutral zone for porcine species and scrotum and testis are located in such a place that the boar is not affected with this temperature. What about the THI, rainfall and light period during summer season? Explain

Ø  You conducted this experiment in summer season; what about winter season? Winter stress is also equally important for sperm production

Ø  Mention the company and country for microscope

Ø  Write the complete composition of Linseed oil ethyl esters and Linseed oil and how differ these two

Ø  What time the semen samples was collected? morning/evening/night, at what time?

Ø  Normally the motility is estimated at 400×; then how did you measure the motility at 200×. Explain

Ø  You write brief methodology for the experimental parameters with suitable references

Ø  You described about the measurement of sexual behavior of boars; mention the suitable references    

Ø  The results on motility is very less in both control and treatment than any normal study in porcine species. Why?

Ø  You include total morphological defects or abnormality along with primary and secondary abnormalities.

Ø  The motility was higher in CASA whereas very low in microscopical analysis; explain why  

Ø  Mention the standard value for different CASA parameters; software settings used in the study with use this CACA analyser.

Ø  You mentioned that significance level was set at p<0,05 in materials and methods. Then why did you quote (p < 0,001); explain

Ø  Throughout the manuscript please delete the word significantly. The P-Value is enough. Also stabilize the (p<0.05) make it capital and italic throughout the manuscripts. It is repeated in different editing formats and always put the P-value after the words indicating increased or decreased. Please use simple past tense. Please use one past tense (were significantly increased, were significantly decreased, were increased, was decreased)    

Ø  Line 188-189: Reaction time and false mount were significantly (p < 0,001) less in the Control group as compared to the Experimental group. But in table, this result is reverse; why and explain

Ø  You mentioned the results of CASA parameters; what about interpretation in the discussion section? Explain why and how EELO affect the CASA parameters in porcine species

Ø  You mentioned the mean ± SD, SEM and P value; if you calculate SD, you will know the SEM. You mentioned A, B in superscript, then again P value. Superscript A, B indicates significant difference; delete SD or SEM and P- value

Ø  You studied the semen quality and sexual behavioural profiles in boar in the EELO treated and control group. EELO supplementation in porcine is due to it has antioxidant properties. But you did not measure the antioxidants either in semen or blood or both and same for free radicals or reactive oxygen species. Why explain.

Ø  All the CASA parameters except LIN are significantly higher in the treatment group as compared those in the control group; how? but no explanation in the discussion

Ø  STR and LIN are positively corrected; but in your study, it is reverse; explain why

Ø  You did not mention the minimum standard values for the different CASA parameters for selection of semen in boar; explain

Ø  The main target of Linseed oil ethyl esters and Linseed oil is to protect the membranous structures; but you did not measure the leakage of intracellular contents of sperm; explain

Ø  What about in-vivo or in-vitro fertility rate in EELO treated and control groups?

Ø  English editing is necessary for whole manuscript       

Ø  Old references need to be replaced with latest one

Ø  References like 23 and 24 are incomplete

Ø  Furthermore, a figure with possible mechanisms of action of the EELO on the male reproductive cells and/or tissues could add to the Discussion or Conclusions.

Ø  A figure with the complete experimental protocol on effect of EELO on the semen production and its quality profiles and sex behavioural profiles in the materials and method section.

Ø  Mention the preservation temperature during the experiment

Ø  What was the refusal rate of basal diet in both groups?

Ø  There are many grammatical errors. I suggest the authors to seek professional English language editor to enhance the readability of the manuscript. 

Ø  What are management protocols followed to reduce the summer stress effect in boar in your experiment?

Ø  How your EELO treatment reduce the summer stress and have you measured any summer stress marker profiles in boar in this study? If not, measure and include in this manuscript.  

Ø  All the semen production and its quality profiles and sex behavioural profiles are correlated with scrotal circumference, and testicular biometric profiles and hormone profiles (reproductive as well as metabolic hormones). Therefore, to confirm the effect of EELO on the male reproductive system, you need to study SC and TV, hormones such as FSH, LH, Testosterone, T3, T4, TSH and cortisol.      

Ø  Preliminary seminal parameters such as colour, smell, mass activity, pH are need to be measured and included in the study.

Ø  What percentage the treatment group is differed with the control group for all the experimental parameters?

Ø  Information on effect of EELO on alteration of biochemical profiles in blood and seminal plasma is lacking in the present study; therefore, you have to study.

Ø  You have to study effect of EELO on how many hours maintained the semen quality parameters suitable for insemination in porcine species (incubation study) with control boar  

Ø  Include figures of live and dead, acrosomal integrity, plasma membrane integrity, sperm abnormalities, DNA fragmentation, and apoptotic changes

Ø  In title, remove the full-stop

Ø  Title may be changed as “Dietary supplementation of Linseed oil ethyl esters improves the sexual behavior and seminal parameters in porcine species”

Ø  Rewrite: 21.8℃ to 21.8 °C

Ø  Rewrite: >50 × 108 total sperm cells per ejaculate to >50 × 108 total sperm cells per ejaculate.

Ø  Rewrite: In the present study twelve Polish Line 990 breed boars ranging between 20 to 26 to In the present study, twelve Polish Line 990 breed boars ranging between 20 to 26

Ø  Rewrite: area of 9 m2 to area of 9 m2

Ø  Rewrite: at 7:00 a.m. and 1:30 121 p.m. to at 0700 h and 1300 h

Ø  Rewrite: ad libitum to ad libitum

Ø  Rewrite: In the experimental group each boar had standard feed was top dressed to In the experimental group, each boar had standard feed was top dressed

Ø  Rewrite: presented before by Singh et al. (2022)[9]; Yeste et al. (2011)[19] to presented before by Singh et al. [9]; Yeste et al. [19]

Ø  How many ejaculates were collected in a boar/week and total ejaculates in a boar for the whole experiment

Ø  Data should be presented either mean ± SD or SEM and no need to present in both forms

Ø  Rewrite: 2,08±0,68 to 2.08 ± 0.68 in all the data in the table and in the text

Ø  Rewrite: p < 0.001 to p < 0.001 in the table and in the text

Ø  The data may be presented as 50% as table and 50% as figure.

Ø  Write full: model, version, year and country for the Statistica v.13.3.

Ø  Statistical analysis is incomplete and write in full and complete

Ø  Discussion needs to be re-written based on the results and interpretation is poor.

Ø  References need to be re-checked with the text

Ø  What significance of this study in the field application? Write

Ø  Mention the limitations of the study and scope for further study with use these result

Ø  Mention the Latitude, Longitude and height from mean sea level of the experimental location

Ø  The data presented in the manuscript is not sufficient to prove the effect of EELO on breeding soundness analysis in boar. There are many information lacking to prove it. Therefore, it may be recommended to study all the other leftover parameters and incorporate in the manuscript.

Author Response

Reviewer 2

Manuscript Title: Linseed oil ethyl esters in boar’s diet improved sexual behaviour and fresh semen quality

Ø  On what basis you selected the dose of Linseed oil ethyl esters (3.0%; 45 mL) and duration of supplementation of 16 weeks? Explain

Fats are a rich source of energy and provide essential polyunsaturated fatty acids (linoleic and linolenic). Pigs make good use of fat of both vegetable and animal origin, and its utilization is determined by the ratio of saturated to unsaturated acids, and the total fat content of the feed, which should not exceed 8-10%.( Zalecenia żywieniowe i wartość pokarmowa pasz dla świń. Normy żywienia świń Eugeniusza R. Greli i Jacka Skomiała Instytut Fizjologii i Żywienia Zwierząt PAN, 2020, wydanie III)

Spermatogenesis in boar testes lasts 40 days, and sperm passage through the epididymis lasts about 10 days (about 7 weeks). The specified values for spermatogenesis and sperm passage through the epididymis are within the range reported for most mammals. Therefore, longer study periods are required to determine the effect of nutrition on permanent changes in semen and sperm quality. Therefore, the 16-week period of EELO supplementation seems to be fully justified in this case. The longer the nutritional studies, the more reliable the results of these studies. (Franc L.R., Avelar G.F., Almeida F.F.L. Spermatogenesis and sperm transit through the epididymis in mammals with emphasis on pigs. Theriogenology (2005) 63  300–318.)

Ø  You have studied only one dose of EELO and it showed beneficial effect. What about the optimal dose?. Therefore, different doses at minimum difference between doses is to be studied to select suitable dose, which should have maximum effect on the reproductive parameters.

This is the preliminary study where the main aim was the possibility of use and evaluation of the effect of dietary linseed oil ethyl esters. The result showed a great opportunity and need further study.

Ø  Have you conducted any comparison study between Linseed oil ethyl esters and Linseed oil on semen production and its quality parameters? On what basis you are claiming the Linseed oil ethyl esters treatment is better than Linseed oil? 

We have not conducted studies to compare the effect of EELO and flaxseed oil on boar semen production and sperm quality parameters. Research by Singh et al. 2022, showed that linseed oil in the diet of boars improves sperm quality parameters, antioxidant status, and the composition of fatty acids in sperm. But EELO are also an excellent source of omega 3 fatty acids, especially α-linoleic acid. Unlike linseed oil, its ethyl esters are devoid of harmful substances, such as amygdalin, linamarin, which damage liver cells. Due to the lower solubility of oxygen in esters, they are more durable and less susceptible to oxidation, epoxidation, peroxidation and gumming processes, and it seems that they can have a positive effect on the body. So we hypothesized that the addition of EELO to the boar’s diet,  could also improve semen quality and boar sperm parameters. Indeed, our study has a limitation because we did not introduce an experimental group with linseed oil in the methodology. However, since the results of our study are promising, it seems advisable to conduct future studies to compare the effects of flaxseed oil and EELO supplementation in the diet on semen quality and boar reproductive performance.

Ø  In this experiment, at least three groups must be included like control (without oil or its ethyl esters), with oil and with oil ethyl esters; then only this experiment will be complete one. 

We agree to reviewer comment, that kind of experimental design would give us broader view but due to permission of Local Ethics Committee and 3R rule we were able to lead experiment on 12 boars. If we decrease the number of animals in group and divide into 3 groups we wouldn’t get reliable results ( sensitivity of the statistical test).

Ø  You conducted this experiment in summer season and mentioned that mean temperature was minimum 21.8. This temperature is fall within thermo-neutral zone for porcine species and scrotum and testis are located in such a place that the boar is not affected with this temperature. What about the THI, rainfall and light period during summer season? Explain

Outside weather conditions: temperature 21.8 min and max +37,9°C, mean humidity was 68%, rain fall was 22mm, light period 1.08.2015 – 15h 17min; 31.08.2015 – 13h 33min

Instytut Meteorologii i Gospodarki Wodnej Państwowy Instytut Badawczy

BULLETIN OF THE NATIONAL HYDROLOGICAL AND METEOROLOGICAL SERVICE August 2015

The animals were kept in controlled conditions without access to paddocks (ASF Regulation of the Minister of Agriculture and Rural Development of August 10, 2021 on measures to be taken in connection with the occurrence of African swine fever)

Ø  You conducted this experiment in summer season; what about winter season? Winter stress is also equally important for sperm production

Season was not included as a research factor (the effect of winter and summer seasons on semen quality ratings). However, it was taken into account for the period of analysis conducted due to the reduced reproductive parameters during the summer season, along with the average temperatures during the study, as stated in the materials and methods. We also exclude here both heat stress and low temperature stress due to the controlled welfare-adapted microclimate conditions prevailing in the experimental piggery where boars were kept. (Council Directive 2008/120/EC of December 18, 2008 laying down minimum standards for the protection of pigs. Ordinance of the Minister of Agriculture and Rural Development of February 15, 2010 on requirements and procedures for the keeping of livestock species for which protection standards have been established by European Union regulations (Journal of Laws No. 56, item 344, as amended).)

Ø  Mention the company and country for microscope

Nikon microscope, Japan

Ø  Write the complete composition of Linseed oil ethyl esters and Linseed oil and how differ these two

Complete composition is shown below. Gas chromatography with mass detector was performed. As you can see are almost similar. Linseed oil ethyl esters are characterized by significantly increased bioavailability, and can be easily absorbed and incorporated into various types of blood lipid fractions. This is due to their undeveloped molecular structure and increased kinetics of free acid release, which contributes to their faster digestion. Due to significantly reduced oxygen solubility, ethyl esters are characterized by much higher durability and stability compared to linseed oil, and are more resistant to oxidation, peroxidation and polymerization processes. The method of obtaining ethyl esters of polyunsaturated fatty acids is covered by patent protection.

Linseed oil

Ret.Time

Area

Area%

 compounds

29,295

2869985

4,95

C16:0

34,082

1847975

3,19

C18:0

34,778

8463708

14,61

cis-C18:1

34,955

449676

0,78

trans-C18:1

36,165

8576965

14,8

cis,cis-C18:2

37,649

35734899

61,67

cis,cis,cis-C18:2

EELO

Ret.Time

Area

Area%

compounds 

30,34

2362341

4,98

C16:0

35,045

1988892

4,19

C18:0

35,62

9037910

19,03

cis-C18:1

35,796

382184

0,8

trans-C18:1

36,803

8219828

17,31

cis,cis-C18:2

38,115

25489775

53,68

cis,cis,cis-C18:2

Ø  What time the semen samples was collected? morning/evening/night, at what time?

Morning before feeding about 6.00 am

Ø  Normally the motility is estimated at 400×; then how did you measure the motility at 200×. Explain

200× magnification, was given in the manuscript by mistake, indeed motility was estimated at 400× magnification.

Ø  You write brief methodology for the experimental parameters with suitable references

Ejaculate volume was determined after the gel fraction was discarded in a graduated tube (mL). Sperm concentration in the ejaculate was determined photometrically using a SpermaCue Porcine photom-eter (Minitube, GmbH, Tiefenbach, Germany) at a light length of 0.7 µm (Kondracki et al. 2021).  Kondracki  S., Iwanina M., Wysokińska A., Banaszewska D., Kordan W., Fraser L., Rymuza K., Górski K. The usefulness of sexual behaviour assessment at the beginning of service to predict the suitability of boars for artificial Insemination. Animals 2021, 11, 3341. https://doi.org/10.3390/ani11123341 Inserted in the manuscript text

Ø  You described about the measurement of sexual behavior of boars; mention the suitable references   

(Kondracki et al. 2021, Singh et al. 2022) Inserted in the manuscript text

Ø  The results on motility is very less in both control and treatment than any normal study in porcine species. Why?

Compared to the results of our previous studies, we do not consider that the sperm motility in fresh boar semen in our study is lower than in any normal study in porcine species (Bielas et al. 2017). Taking into account the results from the experimental group of the current studies, the opposite is true, especially in the case of VAP and VSL. Also compared to the results of boar sperm motility obtained by other researchers, for example Broekhuijse et al. 2012, the results on motility in our results do not appear to be significantly different and lower (Broekhuijse M.L.W., Sostaric E., Feitsma H, Gadella B.M. Application of computer-assisted semen analysis to explain variations in pig fertility. J. Anim. sci. 2012. 90:779–789) http://dx.doi.org/10.2527/jas.2011-4311. Device type, settings, and the algorithm that reproduces the sperm trajectory during the motility assessment may also be considered as possible reasons for differences in CASA system scores and in the obtained results.

Ø  You include total morphological defects or abnormality along with primary and secondary abnormalities.

Sperm morphology were examined for the primary and secondary defects of morphology. Primary defects of morphology: proximal droplet head, anomalies, acrosome damaged, midpice damage, coil tail. Secondary defects of morphology: distal droplet, bent tail, tail-less head, tail-loop (Blom E. 1973). Inserted in the manuscript text

Ø  The motility was higher in CASA whereas very low in microscopical analysis; explain why 

There is always a difference in the assessment of sperm motility performed by CASA and man. In our case, the same subjects assessed subjective mobility under a light microscope. Apparently, their unintentional, erroneous, but research-justified assessment resulted in obtaining such results.

Ø  Mention the standard value for different CASA parameters; software settings used in the study with use this CACA analyser.

Settings of the IVOS were the following: frame acquired 45, frame rate 60 Hz, minimum cell contrast 46, minimum cell size 7, straightness threshold 45%, path velocity threshold 45 μ/s, path velocity cut off 20 μ/s, straight line velocity cutoff 5 μ/s, head size non-motile 7, head intensity non-motile 50, static head size 0.65–4.90, static head intensity 0.50–2.50, static elongation 0–87. Six fields randomly selected by a computer were analyzed for each semen sample. In each sample, 800–1000 sperm cells were evaluated. Inserted in the manuscript text

Ø  You mentioned that significance level was set at p<0,05 in materials and methods. Then why did you quote (p < 0,001); explain

The significance set at p<0,05 was made by the mistake correct sentence is :Data are presented as mean value, SEM and significance level was set at p<0,001.

Ø  Throughout the manuscript please delete the word significantly. The P-Value is enough. Also stabilize the (p<0.05) make it capital and italic throughout the manuscripts. It is repeated in different editing formats and always put the P-value after the words indicating increased or decreased. Please use simple past tense. Please use one past tense (were significantly increased, were significantly decreased, were increased, was decreased)   

According to review manuscript was changed.

Ø  Line 188-189: Reaction time and false mount were significantly (p < 0,001) less in the Control group as compared to the Experimental group. But in table, this result is reverse; why and explain

Reaction time was less and the number of false mounts were lower (p < 0,001) in the Experimental group as compared to the Control  group. (The corrected sentence was inserted into the manuscript text)

Ø  You mentioned the results of CASA parameters; what about interpretation in the discussion section? Explain why and how EELO affect the CASA parameters in porcine species

In this study, all fresh semen quality parameters and CASA attributes (except Linearity) were significantly improved in boars supplemented with EELO in the feed. Although there are conflicting reports on the effect of n-3 PUFA in the diet on the semen properties of males of different species, it seems that supplementation of these compounds affects the cholesterol ratio in the sperm cell membranes, and thus the expression of receptors  and synthesis of sex hormones [Permul et al., 2019]. In addition, a diet rich in n-3 PUFA (as is in the case with EELO) may increase the fluidity of the sperm cell membrane, and thus improve the quality of preserved semen [Tran et al. 2016].

Perumal P, Chang S, Khate K, Vupru K, Bag S. Flaxseed oil modulates semen production and its quality profiles, freezability, testicular biometrics and endocrinological profiles in mithun. Theriogenology 2019;136:47-59.

Tran LV, Malla BA, Sharma AN, Kumar Sachin, Nitin Tyagi, Tyagi AK. Effect of omega-3 and omega-6 polyunsaturated fatty acid enriched diet on plasma IGF-1 and testosterone concentration, puberty and semen quality in male buffalo. Anim Reprod Sci 2016;173:63e72.

Ø  You mentioned the mean ± SD, SEM and P value; if you calculate SD, you will know the SEM. You mentioned A, B in superscript, then again P value. Superscript A, B indicates significant difference; delete SD or SEM and P- value

According to review ± SD was deleted.

Ø  You studied the semen quality and sexual behavioural profiles in boar in the EELO treated and control group. EELO supplementation in porcine is due to it has antioxidant properties. But you did not measure the antioxidants either in semen or blood or both and same for free radicals or reactive oxygen species. Why explain.

Lipid peroxidation is the chain of reactions of oxidative degradation of lipids. In this process free radicals (such as oxyl radicals, peroxyl radicals, and hydroxyl radicals) take electrons from the lipids in cell membranes and subsequently produce reactive intermediates that can undergo further reactions, in the end resulting in cell damage.

Especially it affects polyunsaturated fatty acids, because they contain multiple double bonds in between which lie methylene bridges (-CH2-) that possess very reactive hydrogen atoms. The chemical products of this oxidation are known as lipid peroxides or lipid oxidation products (LOPs).

By measuring live spermatozoa without lipid peroxidation using PI-/BODIPY- (PI - propidium iodide; BODIPY - fluorescent membrane probe) we manage to analyze whether EELO can act against Reactive oxygen species in case of lipid peroxidation. Which is very important because under some circumstances it can lead to destroy DNA, proteins, and enzyme activity as well as acts as molecular to activate signaling pathways initiating cell death (i.e. ferroptosis).

Thanks to the use of BODIPY dye in our experiment it was possible to monitor peroxidative damage of boars’ sperm as a result of lipid peroxidation. When incorporated into cell membranes, this probe is attacked by reactive oxygen metabolites, and by changing its color from red to green, it indicates the exposure of phospholipids to reactive oxygen species and thus lipid peroxidation can be quantified. The target of flow cytometry protocols in sperm assessment can also be lipid peroxidation degradation products such as 4-hydroxynonenal (4-HNE), acrolein (ACR) and malondialdehyde (MDA) [Boe-Hansen et.al, 2019].  Boe-Hansen G.B., Satake N. An update on boar semen assessments by flow cytometry and CASA. Theriogenology (2019) 137, 93-103. doi.org/10.1016/j.theriogenology.2019.05.043

Ø  All the CASA parameters except LIN are significantly higher in the treatment group as compared those in the control group; how? but no explanation in the discussion

The explanation has already been given above.

Ø  STR and LIN are positively corrected; but in your study, it is reverse; explain why

It is difficult for us to explain why, in our study, the linearity of sperm motility in CASA assessment (as opposed to the straightness) is not positively correlated.  In our earlier studies, we observed an increase in the percentage of LIN as the storage time of liquid-conserved semen increased [Bielas et al. 2017]. In contrast, in the current experiment, we observed a bit lower percentage of LIN values in the Experimental group, perhaps this is related to the generally poorer quality of semen in the Control group.

Ø  You did not mention the minimum standard values for the different CASA parameters for selection of semen in boar; explain

In order to select ejaculates, the following minimum standard values of CASA parameters were establshed: PMOT%: (VAP> 45µm/s and STR > 45%), RAPID%: (VAP > 45 µm/s). (The sentence was inserted into the manuscript text)

Ø  The main target of Linseed oil ethyl esters and Linseed oil is to protect the membranous structures; but you did not measure the leakage of intracellular contents of sperm; explain

It seems to us that flow cytometry is a test that, by looking into the structure of the cell membrane, can indirectly determine the degree of leakage of intracellular content from spermatozoa. It seems to us that flow cytometry is a test that, through insight into the structure of the cell membrane, can indirectly determine the degree of leakage of intracellular content from spermatozoa. Therefore, the flow cytometry analysis used in our studies (assessment of cell viability, acrosome and cell membrane integrity, apoptotic and capacitance changes, and membrane lipid peroxidation) can also provide insight into the degree of cell membrane damage, and thus inform about leakage of sperm content.

Ø  What about in-vivo or in-vitro fertility rate in EELO treated and control groups?

We want to emphasize that our research is at a preliminary level, where the main aim was only to evaluate the effect of dietary linseed oil ethyl esters on sexual behaviour and quality parameters of fresh boar semen. The obtained results indicate the need for further research not only in order to assess in vivo or in vitro fertility in boars, but also to assess the manifestations of many important biological functions of their organisms.

Ø  English editing is necessary for whole manuscript      

English editing was improved

Ø  Old references need to be replaced with latest one

According to review manuscript was changed.

Ø  References like 23 and 24 are incomplete

References were completed.

  1. Kołodziej, H.A., Vogt, A., Strzelecki S., Fałat J., Sowa A.E. Sposób wytwarzania estrów alkilowych wyższych kwasów tłuszczowych. Patent, 2014, PL 216194 B1
  2. Kołodziej H.A., Vogt A., Strzelecki S., Steinmetz G.S. Sposób wytwarzania estrów etylowych lub metylowych wyższych kwasów tłuszczowych oraz instalacja do realizacji tego sposobu.Patent, 2012, PL 211325 B1

Ø  Furthermore, a figure with possible mechanisms of action of the EELO on the male reproductive cells and/or tissues could add to the Discussion or Conclusions.

In order to add mechanisms of action of the EELO on the male reproductive cells and/or tissues first of all it needs to be explored in detail so unfortunately, at this stage of research we are not able to specify it and we do not want to speculate.

Ø  A figure with the complete experimental protocol on effect of EELO on the semen production and its quality profiles and sex behavioural profiles in the materials and method section.

Adding an experimental scheme has no substantive basis because the experiment was based on 2 groups of control and with EELO and semen testing. In our opinion, such a scheme does not add anything valuable or meaningful to the materials and methods section. The animals were kept under controlled conditions and such a scheme would make sense if the boars included in the study had access to paddocks or were kept outdoors.

Ø  Mention the preservation temperature during the experiment

The aim of our research was only to assess the effect of EELO supplementation on sex drive and the quality of fresh but not stored semen. Immediately after collection, within 5 minutes, the semen was transported to the laboratory in thermally insulated vessels. In the laboratory, the temperature ranged from 18 to 22 degrees Celsius (room temperature).

Ø  What was the refusal rate of basal diet in both groups?

Feed palatability was not the focus of our study, but from observations of the study population, the addition of oil and EELO translated significantly into feed intake. Thus, the amount of underfeeding was very negligible therefore not included in the study. With such a small population, feed was hand-fed, which translated into the accuracy of observations of underfeeding in the animal population.

Ø  There are many grammatical errors. I suggest the authors to seek professional English language editor to enhance the readability of the manuscript.

English editing was improved

Ø  What are management protocols followed to reduce the summer stress effect in boar in your experiment?

Boars are kept in welfare conditions in accordance with Council Directive 2008/120/EC of December 18, 2008 establishing minimum standards for the protection of pigs. Ordinance of the Minister of Agriculture and Rural Development of February 15, 2010 on the requirements and procedures for the maintenance of livestock species for which the standards of protection are laid down in the regulations of the European Union (Journal of Laws No. 56, item 344 as amended). and no heat stress was observed in the experiment. The physiological and biological parameters of the boars were normal. Therefore, no zootechnical treatments were introduced to level the aforementioned phenomenon.

Ø  How your EELO treatment reduce the summer stress and have you measured any summer stress marker profiles in boar in this study? If not, measure and include in this manuscript. 

The enclosed buildings and facilities where the animals were kept effectively protected the boars from the high outside temperature, keeping heat stress to a minimum. Since no heat stress was observed, no modifications or analyses were made to minimize the effect of high temperatures on the results obtained. We were aware that heat stress would interfere with the evaluation of semen quality and could significantly alter the effect of the dietary supplement on boar semen quality.

Ø  All the semen production and its quality profiles and sex behavioural profiles are correlated with scrotal circumference, and testicular biometric profiles and hormone profiles (reproductive as well as metabolic hormones). Therefore, to confirm the effect of EELO on the male reproductive system, you need to study SC and TV, hormones such as FSH, LH, Testosterone, T3, T4, TSH and cortisol.     

We want to emphasize that our research is at a preliminary level, where the main aim was only to evaluate the effect of dietary linseed oil ethyl esters on sexual behavior and quality parameters of fresh boar semen. Based on the obtained results, further studies are planned, including to assess the effect of EELO supplementation on the quality of semen preserved in a liquid state, frozen semen, oxidative stress, lipid profile and other important parameters, such as scrotal circumference,  testicular biometric profiles and hormone profiles (reproductive as well as metabolic hormones).

Ø  Preliminary seminal parameters such as colour, smell, mass activity, pH are need to be measured and included in the study.

Seminal parameters such as colour, smell, mass activity, pH were also measured as a part of standard preliminary, laboratory assessment. Ejaculates whose aforementioned indicators were not within normal limits were discarded. Only ejaculate with white color, milky consistency, without smell, well mass activity and ph 7.5 were destined for further study. During the study period, only 11 ejaculates that did not meet the assumed criteria of the initial assessment were removed. There were 9 ejaculates from the Control group and 2 ejaculates from the Experimental group.

Ø  What percentage the treatment group is differed with the control group for all the experimental parameters?

The treated group differed from the control group in a significant way for almost all experimental parameters. For example, for the most important parameters of semen and sperm quality, such as: ejaculate volume, concentration per unit of volume and the percentage of spermatozoa with progressive motility, it was 19, 25 and 42%, respectively.

Ø  Information on effect of EELO on alteration of biochemical profiles in blood and seminal plasma is lacking in the present study; therefore, you have to study.

In Materials and Methods section, a sentence was added regarding to blood counts that were conducted to study animal health. Since the supplementation of EELO did not include in the aim of the study the effect on biochemical parameters, adding them to the results section would mislead the recipient and increase the cost of the experiment.

Ø  You have to study effect of EELO on how many hours maintained the semen quality parameters suitable for insemination in porcine species (incubation study) with control boar 

Need further research.

Ø  Include figures of live and dead, acrosomal integrity, plasma membrane integrity, sperm abnormalities, DNA fragmentation, and apoptotic changes

According to  review appropriate figures were added.

Ø  In title, remove the full-stop

The full-stop was removed

Ø  Title may be changed as “Dietary supplementation of Linseed oil ethyl esters improves the sexual behavior and seminal parameters in porcine species”

Title was changed as “Dietary supplementation of Linseed oil ethyl esters improves the sexual behavior and chosen seminal parameters in porcine species”

Ø  Rewrite: 21.8 to 21.8 °C

Improved according to review.

Ø  Rewrite: >50 × 108 total sperm cells per ejaculate to >50 × 108 total sperm cells per ejaculate.

Improved according to review.

Ø  Rewrite: In the present study twelve Polish Line 990 breed boars ranging between 20 to 26 to In the present study, twelve Polish Line 990 breed boars ranging between 20 to 26

Improved according to review.

Ø  Rewrite: area of 9 m2 to area of 9 m2

Improved according to review.

Ø  Rewrite: at 7:00 a.m. and 1:30 121 p.m. to at 0700 h and 1300 h

Improved according to review.

Ø  Rewrite: ad libitum to ad libitum

Improved according to review.

Ø  Rewrite: In the experimental group each boar had standard feed was top dressed to In the experimental group, each boar had standard feed was top dressed

Improved according to review.

Ø  Rewrite: presented before by Singh et al. (2022)[9]; Yeste et al. (2011)[19] to presented before by Singh et al. [9]; Yeste et al. [19]

Improved according to review.

Ø  How many ejaculates were collected in a boar/week and total ejaculates in a boar for the whole experiment

One ejaculate per week was collected from each boar during the 16 weeks of the study, 16 ejaculates were collected from each boar during the 16 weeks of the study, for a total of 192, but only 96 ejaculates were used and evaluated for the study. 

Ø  Data should be presented either mean ± SD or SEM and no need to present in both forms

Improved according to review.

Ø  Rewrite: 2,08±0,68 to 2.08 ± 0.68 in all the data in the table and in the text

Data are now presented as meam and SEM so ±SD was excluded.

Ø  Rewrite: p < 0.001 to p < 0.001 in the table and in the text

Improved according to review.

Ø  The data may be presented as 50% as table and 50% as figure.

Introducing additional figures does not add anything new to the work and, in addition, the reader may miss important details that are included in the description and must be omitted from the diagram. The diagram must not contain many details of the results.

Ø  Write full: model, version, year and country for the Statistica v.13.3.

TIBCO Software Inc. Statistica (Data Analysis Software System)—Version 13. 2017.

Ø  Statistical analysis is incomplete and write in full and complete

The collected results were statistically processed using Statistica v.13.3 (TIBCO Software Inc.). The dataset were analysed for normal distribution by Shapiro–Wilk test.

Which gave evidence that the data tested are not normally distributed.

In order to check the homogeneity of variance Levene test was used.

The Mann–Whitney U test was used to analyse the statistical difference between means of control and treated animals and testing hypothesis of application of EELO on sperm parameters.

Data are presented as mean value, SEM and significance level was set at p<0,001

Ø  Discussion needs to be re-written based on the results and interpretation is poor.

Discussion was improved.

Ø  References need to be re-checked with the text

References were re-checked with the text

Ø  What significance of this study in the field application? Write

In production when reduced semen quality of normospermic boars is observed, it can be a nutritional treatment to improve semen in qualitative and quantitative aspects

Ø  Mention the limitations of the study and scope for further study with use these result

Indeed, our study has a limitation because we did not introduce an experimental group with linseed oil in the methodology. However, since the results of our study are promising, it seems advisable to conduct future studies to compare the effects of flaxseed oil and EELO supplementation in the diet on semen quality characteristics and boar reproductive performance.

Ø  Mention the Latitude, Longitude and height from mean sea level of the experimental location

51.11237095713542, 17.062776984562905 Latitude, Longitude

155 height from mean sea level of the experimental location

Ø  The data presented in the manuscript is not sufficient to prove the effect of EELO on breeding soundness analysis in boar. There are many information lacking to prove it. Therefore, it may be recommended to study all the other leftover parameters and incorporate in the manuscript.

This is the preliminary study where the main aim was the possibility of use and evaluation of the effect of dietary linseed oil ethyl esters. Obtained preliminary results gives us basics and opportunities for further study

Round 2

Reviewer 2 Report

The manuscript has been corrected upto the mark. Therefore, the manuscript may be accepted.